# Integrating pharmacogenomics in three Middle Eastern countries' healthcare (Lebanon, Qatar, and Saudi Arabia): Current insights, challenges, and strategic directions

Said El Shamieh[1☯], Rimah Abdullah Saleem[2☯], Dalal Hammoudi Halat[3], Hana M. A. Fakhoury[2], Kholoud Bastaki[4], Mirna Fawaz[5], Ahmed Malki[6], Rajaa Fakhoury[1,2*]

1 Molecular Testing Laboratory, Department of Medical Laboratory Technology, Faculty of Health Sciences, Beirut Arab University, Beirut, Lebanon, 2 Department of Biochemistry and Molecular Medicine, College of Medicine, Alfaisal University, Riyadh, Saudi Arabia, 3 QU Health, Qatar University, Doha, Qatar, 4 Clinical and Pharmacy Practice Department, College of Pharmacy, QU Health, Qatar University, Doha, Qatar, 5 Department of Nursing, Faculty of Health Sciences, Beirut Arab University, Beirut, Lebanon, 6 Department of Biomedical Sciences, College of Health Sciences, QU Health, Qatar University, Doha, Qatar

☯ These authors contributed equally to this work.
* rfakhoury@bau.edu.lb

## Abstract

### Background and objectives

Pharmacogenomics (PGx) leverages genomic information to tailor drug therapies, enhancing precision medicine. Despite global advancements, its implementation in Lebanon, Qatar, and Saudi Arabia faces unique challenges in clinical integration. This study aimed to investigate PGx attitudes, knowledge implementation, associated challenges, forecast future educational needs, and compare findings across the three countries.

### Methods

This cross-sectional study utilized an anonymous, self-administered online survey distributed to healthcare professionals, academics, and clinicians in Lebanon, Qatar, and Saudi Arabia. The survey comprised 18 questions to assess participants' familiarity with PGx, current implementation practices, perceived obstacles, potential integration strategies, and future educational needs.

### Results

The survey yielded 337 responses from healthcare professionals across the three countries. Data revealed significant variations in PGx familiarity and educational involvement. Qatar and Saudi Arabia participants were more familiar with PGx compared to Lebanon (83%, 75%, and 67%, respectively). Participation in PGx-related talks was most prevalent in Saudi Arabia (96%), followed by Qatar (53%) and Lebanon (35%). Key challenges identified included test cost and reimbursement, insufficient physician knowledge, and lack of infrastructure. Lebanon

**Data availability statement:** The datasets generated and/or analysed during the current study are available in the DRYAD repository: (DOI): https://doi.org/10.5061/dryad.b2rbnzsrb.

**Funding:** The author(s) received no specific funding for this work.

**Competing interests:** The authors have declared that no competing interests exist.

reported the highest concern for test costs (16%), compared to the lowest in Saudi Arabia (5%). Despite these challenges, a strong consensus emerged on PGx's potential to improve patient outcomes, with over 86% of respondents in all three countries expressing this belief. Educational interest areas varied by country, with strong interest in PGx for cancer chemotherapy in Saudi Arabia and Lebanon and for diabetes mellitus in Qatar.

## Conclusion

This study highlights the significant influence of varied educational backgrounds and infrastructural limitations on PGx implementation across Lebanon, Qatar, and Saudi Arabia. The findings emphasize the need for targeted strategies in each country to address these distinct barriers. Integrating PGx education into healthcare training programs and clinical workflows could unlock PGx's potential to optimize patient care.

## Introduction

The completion of the human genome sequencing project in 2001 marked a pivotal moment in biomedical research, laying the foundation for the field of pharmacogenomics (PGx) [1]. PGx examines the influence of an individual's genetic makeup on their response to drugs, using DNA signatures to guide treatment and inform drug development and testing [2]. Recent advances in PGx have deepened our understanding of the factors contributing to individual variations in drug response. Genetic variants correlated with drug responses can now be utilized to tailor drug selection and dosing for each patient [3]. As a cornerstone of precision medicine, PGx integrates genomic, environmental, and lifestyle information to inform medical management decisions, providing a more precise approach to disease prevention, diagnosis, and treatment [4–6]. The increasing number of US FDA approvals for precision therapeutics underscores the growing impact of PGx [7]. However, several challenges impede its routine implementation, including modifications of current clinical practices, knowledge gaps, a lack of interest among healthcare professionals, technological hurdles, and cost-effectiveness issues, particularly in developing countries [8–11].

The literature highlights a disparity in PGx research and implementation between low-resource settings and developed economies, each facing unique obstacles and needs [12,13]. There is a significant need for education and training among healthcare providers to effectively interpret and apply PGx data in clinical settings. This includes integrating PGx into medical curricula and continuous professional development programs [14,15]. In Lebanon, efforts in PGx have been progressing, with several pharmacogenetics laboratories established in the past decade [16]. Studies employing next-generation sequencing and other genotyping techniques demonstrate active participation in this field [17–23]. However, an international survey of PGx experts, including Lebanon, indicated that clinical implementation remains challenging despite the availability of PGx tests and a positive attitude towards their applications. Moreover, while PGx education is included in the curricula of Lebanese higher education institutions [24,25], there is still a need for further education to improve foundational knowledge and capitalize on significant opportunities for advancement [26–28].

In contrast, Qatar has made considerable strides in PGx, supported by an advanced healthcare infrastructure, a national genome program, and numerous educational initiatives focused on precision medicine [29]. Research in Qatar encompasses a variety of pharmacogenomic applications, including antipsychotics, antidepressants, immunosuppressants, warfarin, vitamin D deficiency, and type 2 diabetes risk stratification. To support PGx implementation,

Qatar has undertaken numerous educational initiatives, including courses for healthcare providers, conferences, curricular updates, and the establishment of precision medicine committees at institutional and national levels [29–35]. To support PGx implementation, Qatar has undertaken numerous educational initiatives, including courses for healthcare providers, conferences, curricular updates, and the establishment of precision medicine committees at institutional and national levels [29]. Studies indicate that physicians and pharmacists in Qatar possess positive attitudes and are motivated to engage in PGx-related clinical applications and patient education [36].

PGx is an emerging field in Saudi Arabia with significant implications for personalized medicine. However, there is a notable lack of knowledge and understanding of PGx among healthcare professionals, including community pharmacists and physicians [37]. Research shows that many individuals in the local population carry PGx variants, which affect a substantial percentage of medications dispensed annually, highlighting the need for tailored prescription practices [38–40].

Despite the rapid growth of PGx literature and increasing support from regulatory agencies, there remains a gap in the literature regarding its actual status and implementation [41]. Enhancing education and training initiatives to include pharmacogenomics, improving healthcare professionals' expertise, and raising awareness among policymakers about the importance of PGx could leverage future insights for better healthcare applications [42]. Focused exploration of PGx attitudes, knowledge, implementation, and associated barriers and needs in Lebanon, Qatar, and Saudi Arabia would be instrumental in unlocking PGx's full potential in improving patient care and healthcare systems. To our knowledge, a consolidated investigation of these PGx pillars from the perspective of healthcare professionals and specialists has not been previously undertaken. This study aims to examine the status of PGx implementation in Lebanon, Qatar, and Saudi Arabia, identify associated challenges, predict future educational needs, and compare findings across these three countries.

## Methodology

### Ethical statement and informed consent

Ethical approval for this study was obtained from the institutional review boards of the participating universities: Beirut Arab University (2023-H-0153-HS-R-0545), Qatar University (QU-IRB 1995-E/23), and Alfaisal University (IRB-20270). Informed consent was obtained from all participants online, ensuring their confidentiality and the right to withdraw from the study without any consequences. Participants were informed that all collected data would be anonymous and confidential, with only the principal investigator having access to the data. Completing and submitting the survey was considered an agreement to participate.

### Study design

This study utilized a quantitative cross-sectional research design, involving healthcare professionals (pharmacists, nurses, medical laboratory technologists), university academics, and clinicians from Lebanon, Qatar, and Saudi Arabia. Data was collected through a voluntary, anonymous, private survey to gather PGx perspectives from specialists in the three countries. The study was conducted and reported following the Strengthening the Reporting of Observational Studies in Epidemiology (STROBE) guidelines [43].

### Survey instrument

The survey questions were adapted from the previously published manuscript by of Ghaddar et al., with the approval of Pr. Nathalie Zgheib [44]. The instrument comprised 18 questions

divided into several sections. The first section collected demographic information such as age, gender, country of residence, occupation, and years of experience in clinical practice or research. The second section focused on PGx implementation, examining participants' familiarity with the concept, attendance or presentation of educational talks, and integration of pharmacogenetic testing into their practice or research. The third section explored the obstacles faced in implementing PGx, asking participants to identify the top five obstacles from a provided list. The final section inquired about future considerations, including measures to overcome barriers in PGx implementation, belief in the potential of PGx to improve patient outcomes, interest in learning about pharmacogenomics, and specific topics of interest.

Before launching, the survey was pilot-tested with a small group of healthcare professionals to ensure content clarity and relevance. Based on the pilot test, the survey questions were modified, added, or deleted to improve the content. The pilot responses were not included in the final analysis. The survey items were tailored to country-specific practice norms through consultation with experts from Lebanon and Qatar, where data collection began. For instance, the PGx of pain management was added due to its common testing in Qatari patients.

### Data collection

Participants were recruited through purposive sampling, targeting professionals directly or indirectly involved in PGx. PGx opinion leaders or experts in each country were approached personally via direct email or social media and asked to complete and disseminate the survey within their teams. The survey was launched on September 27th, 2023, and data collection ended on December 23rd, 2023, in Lebanon; January 24th, 2024, in Qatar; and June 7th, 2024, in Saudi Arabia. Participants were briefed about the study's aim in the survey's introductory paragraph and directed to respond anonymously through Google Forms. The survey took approximately 4 minutes to complete.

### Statistical analysis

All statistical analyses were performed using SPSS software (version 26; SPSS Inc., Chicago, IL). Descriptive statistics were used to summarize demographic characteristics and survey responses, with categorical variables expressed as frequencies and percentages. Group comparisons were conducted using the chi-square ($\chi^2$) test of independence to assess whether there were significant differences across the study groups; (1) countries, (2) participants' jobs, (3) academic degrees. The measurable variables included demographic characteristics such as age categories, gender, professional roles, and years of professional experience. Familiarity and involvement with PGx were also compared across the study groups, including familiarity with the concept of PGx, attendance or presentation of educational talks on PGx in the past two years, and the integration of PGx into clinical or research practices. Additionally, future interest and educational needs were assessed by comparing beliefs in PGx's potential to improve patient outcomes, levels of interest in learning about PGx, and specific PGx topics of interest. For each measurable variable, contingency tables were created to cross-tabulate responses by study group, and the $\chi^2$ test was applied to assess independence between these variables. The statistical significance threshold was set at $P \leq 0.05$. When expected cell counts in the contingency table were less than 5, Fisher's exact test was used instead to ensure the validity of the results.

## Results

The demographic characteristics of the study participants from Lebanon, Qatar, and Saudi Arabia are detailed in Table 1. The total sample comprised 337 participants: 145 (43%) from

**Table 1. Demographic characteristics of the study participants in Lebanon, Qatar, and Saudi Arabia.**

| Demographic characteristics | All (N = 337) | | Lebanon (N = 145) | | Qatar (N = 137) | | Saudi Arabia (N = 55) | |
|---|---|---|---|---|---|---|---|---|
| **Age categories N(%)** | | | | | | | | |
| 18–25 | 103 | (30%) | 95 | (66%) | 6 | (4.5%) | 2 | (4%) |
| 26–35 | 84 | (25%) | 26 | (18%) | 45 | (33%) | 13 | (24%) |
| 36–45 | 105 | (32%) | 15 | (10%) | 58 | (42%) | 32 | (57%) |
| 46–55 | 35 | (10%) | 6 | (4%) | 22 | (16%) | 7 | (13%) |
| > 55 | 10 | (3%) | 3 | (2%) | 6 | (4.5%) | 1 | (2%) |
| **Gender N(%)** | | | | | | | | |
| Male | 118 | (35%) | 49 | (34%) | 41 | (30%) | 28 | (51%) |
| Female | 219 | (65%) | 96 | (66%) | 96 | (70%) | 27 | (49%) |
| **Job N(%)** | | | | | | | | |
| Academic | 106 | (32%) | 33 | (23%) | 34 | (25%) | 39 | (70%) |
| Clinician | 29 | (9%) | 9 | (6%) | 13 | (9%) | 8 | (15%) |
| Healthcare Professional | 202 | (59%) | 103 | (71%) | 90 | (66%) | 8 | (15%) |
| **Academic Degree N(%)** | | | | | | | | |
| PhD | 101 | (30%) | 29 | (20%) | 34 | (25%) | 38 | (69%) |
| MD | 30 | (9%) | 10 | (7%) | 13 | (9%) | 7 | (13%) |
| BS in Pharmacy | 64 | (19%) | 9 | (6%) | 45 | (33%) | 10 | (18%) |
| Bachelor | 142 | (42%) | 97 | (67%) | 45 | (33%) | 0 | |
| **Professional Experience N(%)** | | | | | | | | |
| < 1 year | 62 | (18%) | 39 | (27%) | 17 | (12%) | 6 | (11%) |
| 1–5 years | 115 | (34%) | 72 | (50%) | 34 | (25%) | 9 | (16%) |
| 6–10 years | 57 | (17%) | 12 | (8%) | 27 | (20%) | 18 | (33%) |
| 11–20 years | 74 | (22%) | 17 | (12%) | 42 | (31%) | 15 | (27%) |
| > 20 years | 29 | (9%) | 5 | (3%) | 17 | (12%) | 7 | (13%) |

For continuous variables, values are the arithmetic mean ± SD. Categorical variables are shown as numbers (n) and percentages (%).

n: sample size.

Lebanon, 137 (41%) from Qatar, and 55 (16%) from Saudi Arabia. Age distribution showed a significant skew, with a larger proportion of younger participants (18–25 years) in Lebanon (66%) compared to Qatar (4.5%) and Saudi Arabia (4%). Middle-aged groups (26–45 years) were more prevalent in Qatar, while 57% of respondents from Saudi Arabia were between 36 and 45 years old. Gender distribution was similar across the countries, with females constituting the majority: 66% in Lebanon, 70% in Qatar, and 51% in Saudi Arabia. Academics and clinicians were similarly distributed between Qatar and Lebanon. However, there was a notable disparity in healthcare professionals, with Lebanon having a slightly higher proportion (71% vs. 66% in Qatar, Table 1). In terms of academic degrees, the PhD holders represented 30% of our participants, primarily in Saudi Arabia (69%), with Qatar (25%) and Lebanon (20%) trailing. MD holders were 9%, with distributions of 13% in Saudi Arabia, 9% in Qatar, and 7% in Lebanon. BS in Pharmacy holders made up 19%, dominated by Qatar (33%), followed by Saudi Arabia (18%) and Lebanon (6%). The Bachelor's degree (healthcare professionals), the largest group at 42%, were concentrated in Lebanon (67%) and Qatar (33%), with none in Saudi Arabia. Professional experience varied, with Qatar and Saudi Arabia showing higher representation in the more experienced categories than Lebanon (11–20 years and more than 20 years).

The perspectives on PGx across the three countries regarding beliefs, educational interests, and preferred topics were also analyzed (Table 2). Participants from all three countries

**Table 2. Future directions and interest in pharmacogenomics in Lebanon, Qatar, and Saudi Arabia.**

| | Lebanon | | Qatar | | Saudi Arabia | | $\chi^2$ | P |
|---|---|---|---|---|---|---|---|---|
| **Do you believe that pharmacogenomics has the potential to significantly improve patient outcomes in clinical practice?** | | | | | | | | |
| No | 6 | (4%) | 3 | (2%) | 0 | | 5 | 0.278 |
| Yes | 125 | (86%) | 117 | (86%) | 52 | (94%) | | |
| Undecided | 14 | (10%) | 17 | (12%) | 3 | (6%) | | |
| **How interested would you be in learning about pharmacogenomics?** | | | | | | | | |
| Not Interested | 3 | (2%) | 7 | (5%) | 0 | | 18 | 0.005 |
| Slightly Interested | 27 | (19%) | 29 | (21%) | 2 | (4%) | | |
| Interested | 61 | (42%) | 61 | (45%) | 23 | (42%) | | |
| Very Interested | 54 | (37%) | 40 | (29%) | 30 | (55%) | | |
| **What specific pharmacogenomics topics, if any, would you like to learn about?** | | | | | | | | |
| Cancer chemotherapy | 39 | (27%) | 21 | (15%) | 20 | (36%) | 141 | <0.001 |
| Cardiovascular diseases | 18 | (12%) | 27 | (20%) | 10 | (18%) | | |
| Complex diseases* | 41 | (28%) | 4 | (3%) | 12 | (22%) | | |
| Diabetes Mellitus | 0 | | 34 | (25%) | 0 | | | |
| Immune Suppressants | 0 | | 9 | (7%) | 0 | | | |
| Infectious diseases | 15 | (10%) | 11 | (8%) | 4 | (7%) | | |
| Neurodegenerative disorders | 8 | (6%) | 9 | (6%) | 7 | (13%) | | |
| Pain management | 0 | | 12 | (9%) | 0 | | | |
| Precision vaccines and immunity | 8 | (6%) | 0 | | 0 | | | |
| Psychiatric disorders | 12 | (8%) | 10 | (7%) | 0 | | | |
| Rare diseases | 4 | (3%) | 0 | 0 | 2 | (4%) | | |

*Complex diseases are common diseases caused by a combination of genetic, environmental, and lifestyle factors.

For continuous variables, values are the arithmetic mean ± SD.

Categorical variables are shown as numbers (n) and percentages (%). n: sample size. A $\chi^2$ test of independence was used to test if the differences between Lebanon and Qatar are significantly different.

agreed on the potential of PGx to improve patient outcomes, with no significant differences across the three study groups. Interest in learning about PGx varied significantly among the regions (P < 0.005, Table 2). Saudi Arabia recorded the highest proportion of participants very interested in PGx (55%), compared to Lebanon (37%) and Qatar (29%). General interest was similar in Lebanon (42%) and Qatar (45%), while Saudi Arabia showed slightly lower levels (42%). Preference for specific PGx topics differed across countries (p < 0.001). Cancer chemotherapy emerged as a primary focus, particularly in Saudi Arabia (36%) and Lebanon (27%), with Qatar trailing at 15%. Lebanon showed the highest interest in complex diseases (28%). Cardiovascular diseases attracted significant attention in Qatar (20%) and Saudi Arabia (18%), while Lebanon showed lower interest (12%). Qatar uniquely prioritized diabetes mellitus (25%).

Participants were surveyed for their familiarity with the concept of PGx; in Qatar, 75% of the respondents were familiar with PGx, compared to 67% in Lebanon and 83% in Saudi Arabia (Fig 1A). In Qatar, 47% of respondents had not attended or given any talks on PGx, 8% had attended or given one, 31% had a few, and 14% had many (Fig 1B). Comparatively, 65% of respondents in Lebanon had not participated in any talks, 12% had participated in one, 16% in a few, and 7% in many (Fig 1B). In contrast, in Saudi Arabia, only 4% of respondents had not participated in any talks, 10% had participated in one, 40% in a few, and 36% in many (Fig 1B). Regarding PGx integration, 20% of Lebanon's respondents were not interested in integrating it into clinical practice, 66% considered it, 8% occasionally, and 2% routinely

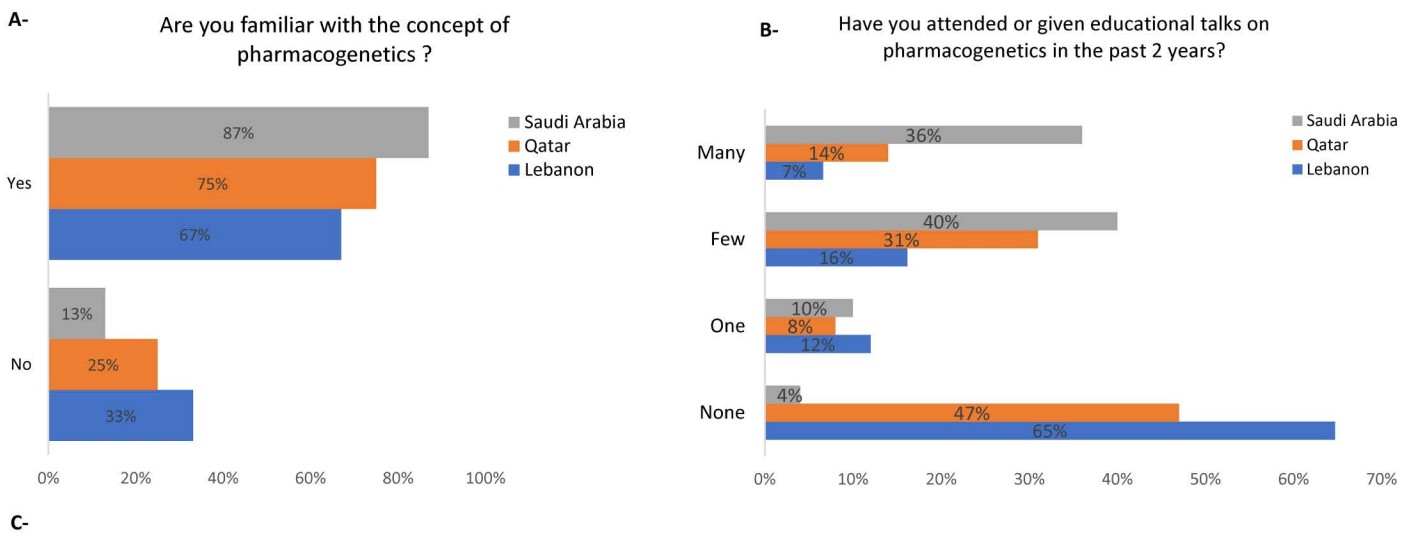

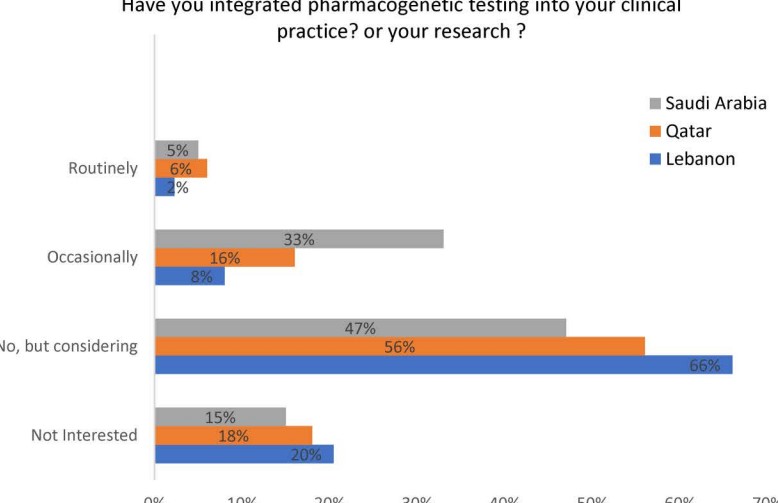

**Fig 1. Knowledge and attitudes towards implementation of pharmacogenomics in clinical practice and research.** A: Familiarity with the Concept of Pharmacogenomics. B. Attendance or Presentation of Educational Talks on Pharmacogenomics in the Past 2 Years. C. Integration of Pharmacogenomics into Clinical Practice or Research Activities.

integrated PGx into their practice or research (Fig 1C). Similar trends were observed in Qatar and Saudi Arabia (Fig 1C).

Fig 2 illustrates the overall challenges faced in PGx implementation. The most significant challenge identified was test cost and reimbursement, highlighted by 13% of respondents. This was followed by insufficient physician knowledge or awareness, indicated by 9% of respondents. The unavailability of test technology, infrastructure, workforce, or experts was also a notable concern, affecting 8% of respondents. About 7% of respondents recognized a lack of funding, investment, or support and insufficient public understanding or awareness. Another prominent issue was unclear or lack of guidelines, cited by 6%. Four percent of respondents found that data from different populations or ethnicities, lack of adequate counselors, adequate communication of results, and limited evidence for clinical utility were significant barriers to PGx implementation in their countries.

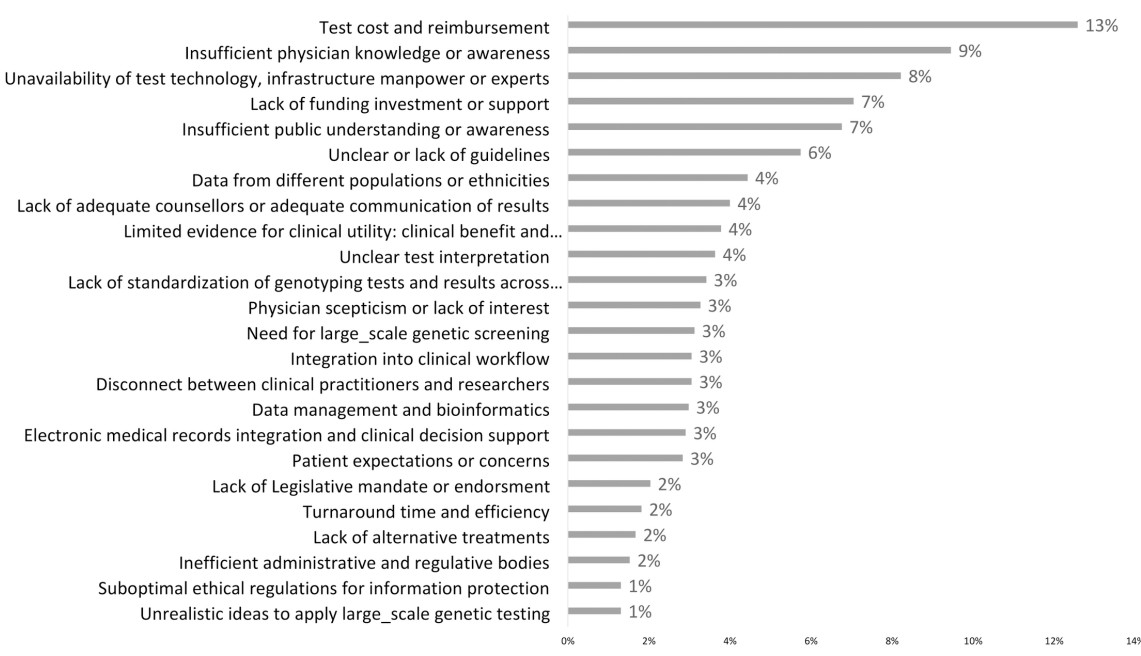

**Fig 2. The overall challenges for bringing pharmacogenetics into clinical practice.**

The PGx challenges revealed significant hurdles, with varying levels of concern across the countries. Regarding test cost and reimbursement, Lebanon reported the highest concern, with 16% of respondents identifying it as a challenge, compared to 10% in Qatar and 5% in Saudi Arabia (Fig 3). Regarding the unavailability of test technology, infrastructure, workforce, or experts, 9% of respondents in Lebanon saw it as a challenge, higher than 6% in Qatar and 4% in Saudi Arabia (Fig 3). Participants from Lebanon and Qatar showed equal concern about insufficient physician knowledge or awareness, with 9% each, notably higher than the 4% in Saudi Arabia (Fig 3). The lack of funding, investment, or support was most pronounced in Lebanon at 9%, compared to 5% in Qatar and 3% in Saudi Arabia (Fig 3). Insufficient public understanding or awareness was identified as a hurdle primarily in Qatar by 8% of participants, followed closely by Lebanon at 7% (Fig 3). In comparison, only 1% of respondents in Saudi Arabia identified it as a challenge. (Fig 3).

A comparative analysis of future trends and interest in PGx based on the job positions of participants is shown in Table 3. When asked if they believe pharmacogenomics has the potential to improve patient outcomes, 91% of academics, 69% of clinicians, and 88% of healthcare professionals responded affirmatively ($\chi^2 = 19$, P = 0.001). In contrast, a small fraction, 1% of academics and 4% of healthcare professionals reacted negatively, with none of the clinicians disagreeing. Additionally, 8% of academics, 31% of clinicians, and 8% of healthcare professionals were undecided. Interest in learning PGx varied across the groups, with 37% of academics, 59% of clinicians, and 44% of healthcare professionals expressing interest, and 49% of academics, 28% of clinicians, and 32% of healthcare professionals being very interested ($\chi^2 = 15$, P = 0.02, Table 3). Only 1% of scholars and 4% of healthcare professionals were uninterested, while no clinicians expressed a lack of interest ($\chi^2 = 15$, P = 0.02). Additionally, 13% of academics, 14% of clinicians, and 20% of healthcare professionals were slightly interested ($\chi^2 = 15$, P = 0.02). The interest analysis according to the different age groups did not show any significant association (S1 Table). Regarding the PGx topics of interest, the responses varied

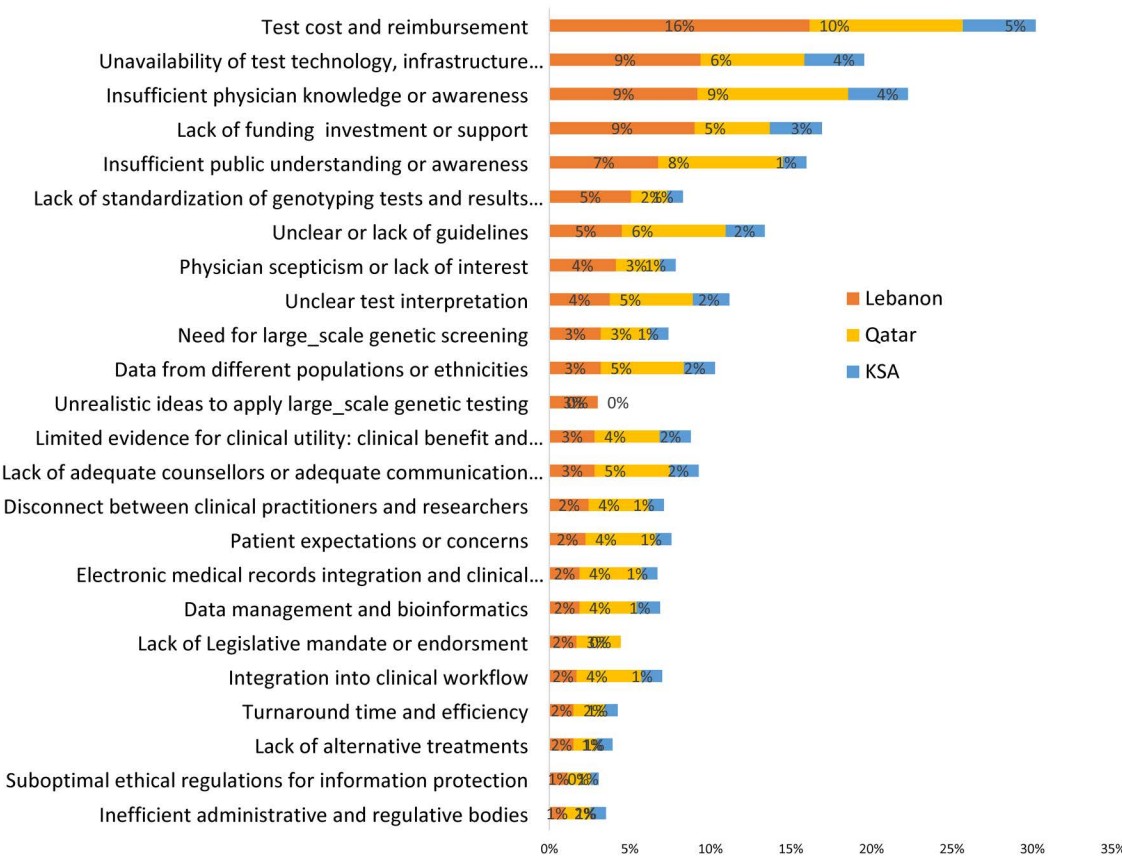

**Fig 3. Challenges for bringing pharmacogenetics into clinical practice in Lebanon, Qatar, and Saudi Arabia.**

but were not significantly different. The three job positions were mainly interested in complex diseases, cancer chemotherapy, cardiovascular diseases, and diabetes mellitus ($\chi^2 = 31$, P = 0.06, Table 3).

Ninety percent of the PhD holders affirmed their belief in PGx and its potential to improve clinical outcomes (P < 0.01, Table 4). The MD participants exhibited 70% affirmative and 30% undecided responses (P < 0.01). The pharmacists had the highest affirmative rate at 97%, with 3% undecided and no negative responses (P < 0.01). In contrast, Bachelor-level Healthcare professionals showed 85% affirmative responses, 10% undecided, and 5% negative (P < 0.01). Among PhD holders, 50% expressed a very high level of interest in learning more about pharmacogenomics (PGx), 36% reported being interested, 13% were slightly interested, and 1% indicated no interest (P < 0.01). MD respondents demonstrated 60% interest, with 27% being very interested, 13% slightly interested, and no disinterest reported (P < 0.01). Pharmacists showed equal proportions of being very interested and interested (42% each), with 16% slightly interested and no reports of disinterest (P < 0.01). Bachelor-level healthcare professionals displayed 44% interest, 28% very interested, 22% slightly interested, and 6% not interested (P < 0.01). In terms of interest in specific PGx topics, PhD respondents reported the highest levels of interest in cancer chemotherapy (27%), complex diseases (18%), and infectious diseases (13%) ($\chi^2 = 41$, P = 0.06). MD participants showed considerable interest in cardiovascular diseases (21%) and cancer chemotherapy (20%), while pharmacists exhibited the highest interest in these two areas (25% and 21% respectively) (Table 4). Bachelor-level

**Table 3. Future directions and Interest in pharmacogenomics according to participants' jobs.**

| | Academic | | Clinician | | Healthcare professional | | $\chi^2$ | P |
|---|---|---|---|---|---|---|---|---|
| **Do you believe that pharmacogenomics has the potential to significantly improve patient outcomes in clinical practice?** | | | | | | | | |
| No | 1 | (1%) | 0 | | 8 | (4%) | 19 | 0.001 |
| Undecided | 9 | (8%) | 9 | (31%) | 16 | (8%) | | |
| Yes | 96 | (91%) | 20 | (69%) | 178 | (88%) | | |
| **How interested would you be in learning about pharmacogenomics?** | | | | | | | | |
| Interested | 39 | (37%) | 17 | (59%) | 89 | (44%) | 15 | 0.02 |
| Not interested | 1 | (1%) | 0 | 0 | 9 | (4%) | | |
| Slightly | 14 | (13%) | 4 | (14%) | 40 | (20%) | | |
| Very Interested | 52 | (49%) | 8 | (28%) | 64 | (32%) | | |
| **What specific pharmacogenomics topics, if any, would you like to learn about?** | | | | | | | | |
| All the complex diseases* | 19 | (18%) | 5 | (17%) | 33 | (16%) | 31 | 0.06 |
| Cancer chemotherapy | 29 | (27%) | 6 | (21%) | 45 | (22%) | | |
| Cardiovascular diseases | 13 | (12%) | 6 | (21%) | 36 | (18%) | | |
| Diabetes Mellitus | 5 | (5%) | 4 | (14%) | 25 | (12%) | | |
| Immune Suppressants | 2 | (2%) | 1 | (3%) | 6 | (3%) | | |
| Infectious diseases | 14 | (13%) | 3 | (10%) | 13 | (6%) | | |
| Neurodegenerative disorders | 12 | (11%) | 0 | | 12 | (6%) | | |
| Pain management | 2 | (2%) | 0 | | 10 | (5%) | | |
| Personalized vaccines and immunity | 4 | (4%) | 0 | | 4 | (2%) | | |
| Psychiatric disorders | 3 | (3%) | 2 | (7%) | 17 | (8%) | | |
| Rare diseases | 3 | (3%) | 2 | (7%) | 1 | (0%) | | |

*Complex diseases are common diseases caused by a combination of genetic, environmental, and lifestyle factors.

For continuous variables, values are the arithmetic mean ± SD. Categorical variables are shown as numbers (n) and percentages (%). n: sample size. A $\chi^2$ test of independence was used to test if the differences between Lebanon and Qatar are significantly different.

healthcare professionals showed notable interest in cancer chemotherapy (21%) and complex diseases (19%), but interest in other topics remained lower across all groups (Table 4).

## Discussion

While the impact of PGx on healthcare is substantial, the realities and expectations of precision medicine vary widely across different contexts and face numerous challenges. This study is the first to investigate the current status of PGx from a specialist viewpoint on a multinational level and is the first in the Arab world to provide a comparative perspective. The results suggest higher familiarity and educational involvement with PGx in Qatar and Saudi Arabia compared to Lebanon. However, a larger proportion of respondents in Lebanon are considering integrating PGx into their practice or research activities, while more respondents in Qatar and Saudi Arabia are already doing so occasionally or routinely.

Significant differences were noted in the level of uncertainty about PGx's potential and specific topics of interest, particularly regarding diabetes mellitus (Qatar) and cancer chemotherapy (Saudi Arabia). These differences are critical for tailoring educational and implementation strategies in each country, underscoring the variability in perceptions and needs for PGx across healthcare systems with different structures. To effectively implement precision medicine in clinical practice, it is crucial to evaluate the specific context of each healthcare system, including the current state of implementation, existing obstacles, and unique challenges.

PGx implementation in research and clinical practice in Arab and Middle Eastern countries faces several documented challenges. One significant challenge is the lag in biomedical

**Table 4. Future directions and Interest in pharmacogenomics according to the academic Degree.**

| | PhD | | MD | | Bachelor in Pharmacy | | Bachelor | | χ² | P |
|---|---|---|---|---|---|---|---|---|---|---|
| No | 1 | (1%) | | | 0 | | 8 | (5%) | 25 | <0.01 |
| Undecided | 9 | (9%) | 9 | (30%) | 2 | (3%) | 14 | (10%) | | |
| Yes | 91 | (90%) | 21 | (70%) | 62 | (97%) | 120 | (85%) | | |
| Interested | 37 | (36%) | 18 | (60%) | 27 | (42%) | 63 | (44%) | 24 | <0.01 |
| Not interested | 1 | (1%) | | | 0 | | 9 | (6%) | | |
| Slightly | 13 | (13%) | 4 | (13%) | 10 | (16%) | 31 | (22%) | | |
| Very Interested | 50 | (50%) | 8 | (27%) | 27 | (42%) | 39 | (28%) | | |
| All the complex diseases* | 18 | (18%) | 5 | (17%) | 5 | (8%) | 29 | (19%) | 41 | 0.06 |
| Cancer chemotherapy | 28 | (27%) | 6 | (20%) | 15 | (21%) | 32 | (21%) | | |
| Cardiovascular diseases | 13 | (13%) | 6 | (21%) | 16 | (25%) | 20 | (14%) | | |
| Diabetes Mellitus | 5 | (5%) | 4 | (13%) | 4 | (6%) | 21 | (14%) | | |
| Immune Suppressants | 2 | (2%) | 1 | (3%) | 3 | (5%) | 3 | (2%) | | |
| Infectious diseases | 13 | (13%) | 3 | (10%) | 4 | (6%) | 10 | (5%) | | |
| Neurodegenerative disorders | 11 | (11%) | 1 | (3%) | 6 | (10%) | 6 | (3%) | | |
| Pain management | 2 | (2%) | | | 5 | (8%) | 5 | (2%) | | |
| Personalized vaccines and immunity | 3 | (3%) | | | 2 | (3%) | 3 | (12%) | | |
| Psychiatric disorders | 3 | (3%) | 2 | (7%) | 5 | (8%) | 12 | (7%) | | |
| Rare diseases | 3 | (3%) | 2 | (7%) | 0 | | 1 | (1%) | | |

*Complex diseases are common diseases caused by a combination of genetic, environmental, and lifestyle factors.

For continuous variables, values are the arithmetic mean ± SD. Categorical variables are shown as numbers (n) and percentages (%). n: sample size. A χ² test of independence was used to test if the differences between Lebanon and Qatar are significantly different.

research output in these countries [45]. Another is the low genomic literacy rates among healthcare professionals, leading to a gap in translating genomic knowledge to patient care [46]. This disparity can hinder the development and application of PGx knowledge, necessitating concerted efforts to bridge this gap [12]. PGx implementation in Lebanon mirrors challenges faced by other nearby countries. Ongoing efforts to raise awareness among healthcare professionals in Lebanon [47] indicate growing interest in this field. However, limited availability, high costs of genetic testing, and delays in translating results underscore the need for solutions to these challenges [47].

PGx challenges varied between countries, highlighting the need for tailored strategies to address specific issues. Developing comprehensive training programs for healthcare professionals, including physicians and pharmacists, can enhance their understanding and utilization of PGx in clinical decision-making [48].

Addressing economic factors in Lebanon would require policy changes to improve accessibility and affordability of PGx testing, potentially through government subsidies or insurance coverage [46,49,50]. In Qatar, where more respondents are already integrating PGx into their practice, developing robust clinical decision support systems and consultation services could facilitate regular use of PGx in patient care.

Several recommendations can be offered based on the study's findings. Firstly, educational and implementation strategies should be tailored to each country's needs and barriers [51]. In Qatar, efforts should focus on transitioning from occasional or routine PGx integration to a more comprehensive and standardized approach. In Lebanon, targeted initiatives should support and incentivize PGx integration into practice or research activities. Addressing economic factors and raising awareness about PGx's potential benefits could encourage adoption in Lebanon and other countries.

In terms of PGx interest areas, specific attention should be given to diabetes mellitus in Qatar and cancer chemotherapy in Saudi Arabia, where there is significant interest due to the high incidence of certain cancers [52–55]. Breast cancer is the most commonly diagnosed cancer among Lebanese and Saudi females, with a significant increase in incidence rates over the years [54,56]. Such high incidence of breast cancer in the two countries may justify the high interest in cancer PGx across participants from Saudi Arabia (36%) and from Lebanon (27%). Our findings are in parallel with global data on PGx of cancer therapies, where the oncology field has been leading on the forefront of precision medicine [57], and almost half of the FDA approvals for pharmacogenomic testing belong to the field of cancer therapeutics [58]. When examining respondents' interest in PGx-specific topics, it becomes evident that certain diseases, such as complex diseases, cardiovascular diseases, and diabetes mellitus, may share overlapping etiology. Nevertheless, their pharmacogenomic implications and therapeutic strategies remain distinct. Internal and family medicine practitioners are often responsible for comprehensively managing complex diseases. In contrast, endocrinologists primarily address the intricate metabolic and hormonal aspects of diabetes mellitus, and cardiologists focus on the hemodynamic and structural complexities of cardiovascular diseases. By distinguishing these categories, we can capture clinicians' diverse and specialty-specific interests in PGx topics.

Collaboration with stakeholders, including healthcare professionals, educators, policymakers, and patients, is essential for successful PGx implementation [59]. Engaging primary care physicians and pharmacists can enhance their knowledge and experience with PGx testing and facilitate its integration into routine clinical practice [60]. The importance of preemptive genotyping for precision medicine should be emphasized, as it can optimize medication selection and dosing, improving healthcare outcomes [61].

Real-life examples of successful pharmacogenomic implementation include preemptive genotyping at institutions such as Vanderbilt and St. Jude Children's Research Hospital, which provides a basis for overcoming implementation challenges [62].

The stratified analysis according to the academic degree, showed interesting patterns (Table 4). Almost all PhD holders, pharmacists, and most MDs and bachelor-level healthcare professionals (>70%) believed that PGx could improve clinical outcomes. Interest in learning PGx varied significantly, with 50% of the PhDs being very interested, 60% of the MDs interested, and 42% and 28% of the pharmacists and bachelors (respectively) being very interested. Topic preferences were cancer chemotherapy (highest across groups), complex diseases (notable for PhDs and Bachelors), and cardiovascular diseases (preferred by MDs and Pharmacists).

Many clinicians can also have teaching activities; thus, their specialty areas might overlap with those of academics. We have carefully checked if the enrolled clinicians were also academics. Interestingly, almost all of them, except two from Lebanon, had teaching activities, and they specialize in PGx/Personalized Medicine and Medical Genetics. Interestingly, our academics had different specialties, hindering the need to analyze the data according to specialty area.

While the current results provide valuable insights into the differences between the three surveyed countries regarding familiarity, educational involvement, and challenges in integrating PGx into their healthcare systems, it is essential to highlight the limitations. First, the study's findings are based on self-reported data that can be subject to recall bias and social desirability bias, potentially influencing the accuracy of the responses. Second, the sample size and representativeness of respondents in the three countries should be considered. Third, the study may not have fully captured healthcare professionals' diverse perspectives and experiences in different practice settings, including those in rural or smaller healthcare facilities. Fourth, the survey did not gauge data on participant's awareness of existing pharmacogenomic guidelines; such data would have provided valuable insights into their familiarity with standardized practices.

This study underscores the crucial need for education and training in PGx, as the lack of awareness and expertise among healthcare professionals can pose a significant barrier to integrating PGx information into clinical decision-making processes. Arab and Middle Eastern nations face multifaceted obstacles in advancing PGx research and integrating it into clinical practice. These challenges encompass resource constraints, educational gaps, regulatory hurdles, ethical considerations, data limitations, and economic factors. A comprehensive understanding of these challenges is crucial for devising effective strategies to overcome them.

## Supporting information

**S1 Table. Interest level of the participants according to the age groups.** Categorical variables are shown as numbers (n) and percentages (%). n: sample size. A $\chi^2$ test of independence was used to test if the differences between Lebanon and Qatar are significantly different. (DOCX)

## Author contributions

**Conceptualization:** Said El Shamieh, Dalal Hammoudi Halat, Kholoud Bastaki, Ahmed Malki, Rajaa Fakhoury.

**Formal analysis:** Said El Shamieh, Dalal Hammoudi Halat.

**Investigation:** Said El Shamieh, Rimah Abdullah Saleem, Hana M. A. Fakhoury, Kholoud Bastaki, Mirna Fawaz.

**Methodology:** Said El Shamieh, Hana M. A. Fakhoury.

**Project administration:** Said El Shamieh, Dalal Hammoudi Halat, Hana M. A. Fakhoury, Ahmed Malki, Rajaa Fakhoury.

**Resources:** Mirna Fawaz, Rajaa Fakhoury.

**Supervision:** Said El Shamieh, Hana M. A. Fakhoury, Kholoud Bastaki, Rajaa Fakhoury.

**Writing – original draft:** Said El Shamieh, Rimah Abdullah Saleem, Dalal Hammoudi Halat, Hana M A Fakhoury.

**Writing – review & editing:** Said El Shamieh, Rajaa Fakhoury.

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
