## [Decision Letter · Decision Letter 0]

2 Dec 2024

PONE-D-24-33568Integrating Pharmacogenomics in Middle Eastern Healthcare: Current Insights, Challenges, and Strategic DirectionsPLOS ONE

Dear Dr. Fakhoury,

Thank you for submitting your manuscript to PLOS ONE. After careful consideration, we feel that it has merit but does not fully meet PLOS ONE’s publication criteria as it currently stands. Therefore, we invite you to submit a revised version of the manuscript that addresses the points raised during the review process.

Contradicting comments were provided by the two reviewers. Although the data and the findings are interesting and of importance, pertinent comments were raised especially by reviewer#2, hence a major revision is recommended.

Authors are required to revise the manuscript accordingly and resubmit. 

We look forward to receiving your revised manuscript.

Kind regards,

Hoh Boon-Peng, PhD

Academic Editor

PLOS ONE

Journal Requirements:

Reviewers' comments:

Reviewer's Responses to Questions

**Comments to the Author**

1. Is the manuscript technically sound, and do the data support the conclusions?

Reviewer #1: Yes

Reviewer #2: Partly

2. Has the statistical analysis been performed appropriately and rigorously? 

Reviewer #1: Yes

Reviewer #2: No

3. Have the authors made all data underlying the findings in their manuscript fully available?

Reviewer #1: Yes

Reviewer #2: Yes

4. Is the manuscript presented in an intelligible fashion and written in standard English?

Reviewer #1: Yes

Reviewer #2: Yes

5. Review Comments to the Author

Reviewer #1: Dear Authors,

I only have very few question/recommendation.

1. Do you have the data on the level of eduction for the subjects? (eg. PhD, Masters, Bsc)

would be interesting to see if there is a difference in acceptance towards Pgx between each group

2. Would be great if you also have data on their awareness of any guidelines related to Pgx.

3. Could you elaborate more on the level of interest (for gruops who are interested to know more) with age?

4. Is there data on pgx practice in Lebanon, Qatar, and Saudi Arabia, and does it collaborate with the level of interest for each respondent grouped by countries?

Reviewer #2: Reviewer Comments:

1. Title Appropriateness: The title of the manuscript is not suitable as it refers to the Middle East, which comprises 18 countries. However, the study is limited to three countries: Lebanon, Qatar, and Saudi Arabia. The data presented may not adequately represent the healthcare population of the entire region. I recommend revising the title to “Integrating Pharmacogenomics in Three Middle Eastern Countries’ Healthcare (Lebanon, Qatar, and Saudi Arabia): Current Insights, Challenges, and Strategic Direction” to more accurately reflect the scope of the research.

2. Data Sufficiency: The results and areas of coverage presented in the manuscript are insufficient for publication in a Q1 journal. To enhance the study's contribution to the field, I suggest providing a broader dataset or deeper insights into the healthcare practices and pharmacogenomics landscape within the selected countries.

3. Clarification of Roles: The roles of academics, healthcare professionals, and clinicians appear to overlap without clear distinctions. The manuscript should specify the areas of specialty for the academic participants, as both clinicians and healthcare professionals may also serve as academicians. Clarifying these roles will strengthen the manuscript's clarity and precision.

4. Statistical Analysis Issues: The statistical analysis presented is inadequate. For example, the statement, “A χ2 test of independence was used to test if the differences between Lebanon and water are significantly different,” is confusing and lacks clarity. It does not provide a significance value for the measurable variables, which is essential for interpreting the results. I recommend revising this section to include clear explanations of the statistical methods used and the significance of the findings.

5. Overlap in Pharmacogenomic Specifications: There are overlapping specifications regarding pharmacogenomics topics. For instance, conditions such as diabetes mellitus and cardiovascular diseases may be influenced by a combination of genetic, environmental, and lifestyle factors, which are similar to the characteristics of complex diseases. It is important to delineate these topics clearly to avoid confusion and enhance the manuscript’s focus.

6. Discussion Quality: The discussion section is well-written and effectively outlines the challenges faced in the integration of pharmacogenomics, along with proposed strategies to overcome these obstacles. This part of the manuscript is commendable and adds significant value to the overall narrative.

6. PLOS authors have the option to publish the peer review history of their article (what does this mean? ). If published, this will include your full peer review and any attached files.

**Do you want your identity to be public for this peer review?** For information about this choice, including consent withdrawal, please see our Privacy Policy .

Reviewer #1: **Yes: ** Fazleen Haslinda Mohd Hatta

Reviewer #2: No

---

## [Author Response · Author response to Decision Letter 1]

18 Dec 2024

Reviewer #1:

1. Do you have the data on the level of education for the subjects? (eg. PhD, Masters, Bsc)

would be interesting to see if there is a difference in acceptance towards Pgx between each group

We thank the reviewer for the pertinent comment. Accordingly, we have added all our participants' academic degrees (Table 1, page 20). We have described these results (lines 177-182, page 8) and then compared the difference in acceptance towards PGx in each group (Table 4, page 23).

This analysis showed interesting patterns that we describe in the Results section (lines 247-261, page 10 and lines 262-263, page 11) and the Discussion section (line 336-342, page 14).

2. Would be great if you also have data on their awareness of any guidelines related to Pgx.

We thank the reviewer for this important comment. Unfortunately, we don’t have this data and we have added this to the limitations section (line 357-359, page 14)

3. Could you elaborate more on the level of interest (for groups who are interested to know more) with age?

We thank the reviewer for highlighting this issue. We analyzed the interest level according to the different age groups and found no significant associations (Supplementary Table). We also describe this in the results section (lines 241-242, page 10).

4. Is there data on pgx practice in Lebanon, Qatar, and Saudi Arabia, and does it collaborate with the level of interest for each respondent grouped by countries?

We thank the reviewer for this comment. Data on pharmacogenomics practice in the three countries exists to some extent in literature, and this was highlighted in the Introduction section with relevant references: references 16-27 for Lebanon, 29-36 for Qatar, and 37-40 for Saudi Arabia. Yes, the level of interest varied across the three countries, as shown in Table 2 (page 21). We have described all the details in the results section (lines 186-196, page 8).

Reviewer #2: Reviewer Comments:

1. Title Appropriateness: The title of the manuscript is not suitable as it refers to the Middle East, which comprises 18 countries. However, the study is limited to three countries: Lebanon, Qatar, and Saudi Arabia. The data presented may not adequately represent the healthcare population of the entire region. I recommend revising the title to “Integrating Pharmacogenomics in Three Middle Eastern Countries’ Healthcare (Lebanon, Qatar, and Saudi Arabia): Current Insights, Challenges, and Strategic Direction” to more accurately reflect the scope of the research.

We thank the reviewer for pointing out this issue. We have used the reviewer’s suggestion for the title of our article.

2. Data Sufficiency: The results and areas of coverage presented in the manuscript are insufficient for publication in a Q1 journal. To enhance the study's contribution to the field, I suggest providing a broader dataset or deeper insights into the healthcare practices and pharmacogenomics landscape within the selected countries.

We thank the reviewer for being keen about data coverage presented in the manuscript. Indeed, a broader dataset would be beneficial suggestion, and providing it would be interesting for ongoing research. For the current study, we exhausted different means of contacting pharmacogenomics experts in the three countries, by approaching them personally, given their known expertise in the field and their reputation. We also sought their support in disseminating the survey among their teams. We reached to broader pharmacist communities to seek specialist participation. Given that the primary authors from each country are academicians, we used our contacts among specialized academicians and researchers to get more engagement in the survey. However, we need to appreciate the relative novelty of the field in the region and the fact that Qatar and Lebanon are both small countries. Finding specialists within the pharmacogenomics domain whose participation adds value to the study should be endeavored carefully. Nevertheless, we have accounted for sample size as one of the study limitations in the Discussion section.

To satisfy the reviewer's concerns and enhance the study contribution to the field, we provided deeper insights into our data by additional analysis on academic degrees, interests in pharmacogenomics, and age. The analysis according to the academic degree showed significantly different patterns as the interest for PGX varied depending on the academic degree (PhD vs. MD vs. BS in Pharmacy vs. BS in a healthcare field (Results, lines 247-261, page 10 and lines 262-263, page 11, and Table 4, page 23). In contrast, the age of the participants was not associated with the interest levels (lines 241-242, page 10).

3. Clarification of Roles: The roles of academics, healthcare professionals, and clinicians appear to overlap without clear distinctions. The manuscript should specify the areas of specialty for the academic participants, as both clinicians and healthcare professionals may also serve as academicians. Clarifying these roles will strengthen the manuscript's clarity and precision.

We thank the reviewer for pointing out this pertinent issue. After carefully checking the clinicians enrolled in our study, we noticed that only two (from Lebanon) had teaching activities. The latter clinicians specialized in Pharmacogenomics/Personalized Medicine and Medical Genetics. Interestingly, our academics were all those of these two specialties. As this is a critical point, we have now clarified it in the Discussion section (page 14, line 344-348).

4. Statistical Analysis Issues: The statistical analysis presented is inadequate. For example, the statement, “A χ2 test of independence was used to test if the differences between Lebanon and water are significantly different,” is confusing and lacks clarity. It does not provide a significance value for the measurable variables, which is essential for interpreting the results. I recommend revising this section to include clear explanations of the statistical methods used and the significance of the findings.

We do entirely agree with the reviewer and acknowledge that we didn’t provide initially enough details concerning the statistical analysis section. Accordingly, we have provided all the necessary details in the methods section (lines 142-157, page 6).

5. Overlap in Pharmacogenomic Specifications: There are overlapping specifications regarding pharmacogenomics topics. For instance, conditions such as diabetes mellitus and cardiovascular diseases may be influenced by a combination of genetic, environmental, and lifestyle factors, which are similar to the characteristics of complex diseases. It is important to delineate these topics clearly to avoid confusion and enhance the manuscript’s focus.

We agree with the reviewer’s observation regarding the overlap in some pharmacogenomics topics. In response, we emphasize that including these overlapping categories was a deliberate decision to reflect our respondents' varied interests and areas of expertise. Although these conditions share a common etiology, their pharmacogenomic implications and therapeutic approaches are distinct. For instance, internal medicine and family medicine practitioners typically manage complex diseases. In contrast, endocrinologists predominantly focus on endocrine disorders like diabetes mellitus, while cardiologists specialize in cardiovascular diseases. By delineating these categories, we captured clinicians' variable interest in PGx topics. We have now discussed this point (lines 318-325, page 13).

6. Discussion Quality: The discussion section is well-written and effectively outlines the challenges faced in the integration of pharmacogenomics, along with proposed strategies to overcome these obstacles. This part of the manuscript is commendable and adds significant value to the overall narrative.

We thank the reviewers for the interest they showed in recommending our discussion section and highlighting its value.

Based on the above, we hope to have satisfied the reviewers’ comments and to have the amended version of the manuscript acceptable for publication in PLOS One.

---

## [Decision Letter · Decision Letter 1]

26 Jan 2025

Integrating Pharmacogenomics in Three Middle Eastern Countries’ Healthcare (Lebanon, Qatar, and Saudi Arabia): Current Insights, Challenges, and Strategic Directions

PONE-D-24-33568R1

Dear Dr. Fakhoury,

We’re pleased to inform you that your manuscript has been judged scientifically suitable for publication and will be formally accepted for publication once it meets all outstanding technical requirements.

Kind regards,

Hoh Boon-Peng, PhD

Academic Editor

PLOS ONE

Additional Editor Comments (optional):

Reviewers' comments:

Reviewer's Responses to Questions

**Comments to the Author**

1. If the authors have adequately addressed your comments raised in a previous round of review and you feel that this manuscript is now acceptable for publication, you may indicate that here to bypass the “Comments to the Author” section, enter your conflict of interest statement in the “Confidential to Editor” section, and submit your "Accept" recommendation.

Reviewer #1: All comments have been addressed

Reviewer #2: All comments have been addressed

2. Is the manuscript technically sound, and do the data support the conclusions?

Reviewer #1: Partly

Reviewer #2: Yes

3. Has the statistical analysis been performed appropriately and rigorously? 

Reviewer #1: Yes

Reviewer #2: Yes

4. Have the authors made all data underlying the findings in their manuscript fully available?

Reviewer #1: Yes

Reviewer #2: Yes

5. Is the manuscript presented in an intelligible fashion and written in standard English?

Reviewer #1: Yes

Reviewer #2: Yes

6. Review Comments to the Author

Reviewer #1: This straightforward study offers an overview of the current state of pharmacogenomics (Pgx) in representative Middle Eastern countries. I wholeheartedly support its publication, as it is crucial to advance Pgx and digital health in the region and beyond. Additionally, there is an urgent need to generate more interest in this field to drive further progress.

Reviewer #2: All the areas of concern raised by reviewer have been addressed clearly by the author.

The necessary changes were performed, and appropriate explanations were provided.

7. PLOS authors have the option to publish the peer review history of their article (what does this mean? ). If published, this will include your full peer review and any attached files.

**Do you want your identity to be public for this peer review?** For information about this choice, including consent withdrawal, please see our Privacy Policy .

Reviewer #1: **Yes: ** Fazleen Haslinda Mohd Hatta

Reviewer #2: **Yes: ** Boon-Keat Tan

---

## [Editor Report · Acceptance letter]

PONE-D-24-33568R1

PLOS ONE

Dear Dr. Fakhoury,

I'm pleased to inform you that your manuscript has been deemed suitable for publication in PLOS ONE. Congratulations! Your manuscript is now being handed over to our production team.

Kind regards,

on behalf of

Professor Dr Hoh Boon-Peng

Academic Editor

PLOS ONE